# Recent Progress on Plant Apomixis for Genetic Improvement

**DOI:** 10.3390/ijms252111378

**Published:** 2024-10-23

**Authors:** Lihua Xue, Yingying Zhang, Fang Wei, Gongyao Shi, Baoming Tian, Yuxiang Yuan, Wenjing Jiang, Meiqi Zhao, Lijiao Hu, Zhengqing Xie, Huihui Gu

**Affiliations:** 1Henan International Joint Laboratory of Crop Gene Resources and Improvements, School of Agricultural Sciences, Zhengzhou University, Zhengzhou 450001, China; 18832239815@163.com (L.X.); zzuzyy123@163.com (Y.Z.); fangwei@zzu.edu.cn (F.W.); shigy@zzu.edu.cn (G.S.); tianbm@zzu.edu.cn (B.T.); jiang_wenjing2021@163.com (W.J.); 15776109263@163.com (M.Z.); hu_lijiao@163.com (L.H.); 2Institute of Horticulture, Henan Academy of Agricultural Sciences, Graduate T & R Base of Zhengzhou University, Zhengzhou 450002, China; yuxiangyuan126@126.com; 3School of Life Sciences, Zhengzhou University, Zhengzhou 450001, China

**Keywords:** apomixis, cloned seed, apomixis-related gene, *MiMe*, genetic improvement breeding

## Abstract

Apomixis is a reproductive process that produces clonal seeds while bypassing meiosis (or apomeiosis) without undergoing fertilization (or pseudo-fertilization). The progenies are genetically cloned from their parents, retaining the parental genotype, and have great potential for the preservation of genes of interest and the fixing of heterosis. The hallmark components of apomixis include the formation of female gametes without meiosis, the development of fertilization-independent embryos, and the formation of functional endosperm. Understanding and utilizing the molecular mechanism of apomixis has far-reaching implications for plant genetic breeding and agricultural development. Therefore, this study focuses on the classification, influencing factors, genetic regulation, and molecular mechanism of apomixis, as well as progress in the research and application of apomixis-related genes in plant breeding. This work will elucidate the molecular mechanisms of apomixis and its application for plant genetic improvement.

## 1. Introduction

In conventional genetic breeding, seeds are formed through double fertilization, in which two sperms combine with an egg to produce an embryo and a central cell to form an endosperm (Figure 1a). Seed formation also occurs in asexual reproduction, where the normal process involving meiosis and fertilization is replaced. Agamogenesis refers to a mode that does not rely on gametogamy to produce seeds [1]. Agamogenesis leads to the formation of clonal seeds in a variety of ways, and there are three common developmental components: (1) bypassing meiosis during embryo sac formation (apomeiosis), (2) fertilization-independent embryonic development, and (3) the formation of an endosperm through double-fertilization-independent fertilization or without sperm [2] (Figure 1).

Since agamogenesis produces asexual seeds, the descendants are genetic clones of the mother, preserving the maternal genotypes [4,5], which helps preserve genes of interest through seed reproduction and has great advantages in heterosis fixation and plant breeding. Taking advantage of agamogenesis, breeders can produce new varieties of cloned seeds more quickly at a low cost, and seed companies can accelerate the pace of breed improvement to yield novel and better varieties faster than their competitors. Moreover, growers can save money on hybrid seeds and obtain high yields and income.

Asexual seed production is considered the Holy Grail of plant biology [6]. Mimicking asexual reproduction in crops would provide an effective mechanism for preserving important genotypes and phenotypes in agriculture. Agamogenesis is particularly frequent in some families, such as Gramineae [7,8], Compositae [4,9], Melastomataceae [10,11], and the Rose family [12,13]. According to incomplete statistics, more than 400 genera have been found to possess the ability of asexual reproduction [2]. However, no major seed crop varieties were bred through agamogenesis, and attempts to introduce asexual traits into sexual parent crops through cross-breeding were largely unsuccessful [14]. Therefore, breeders must explore the regulatory mechanisms of plant asexual reproductive development. Today, the development of synthetic biology, systematic biology, and genomics, as well as the rise of modern biological technology, has promoted the technological innovation of double-haploid technology, haploid breeding technology, and genome-editing techniques, such as molecular-marker-assisted breeding technology. The combination of advanced technologies and the apomixis mechanism would greatly promote asexual breeding research and provide new routes for plant genetic improvement.

In recent years, researchers from various countries have carried out relevant studies on agamogenesis in different species. Although their research directions were different, all made certain progress. In this review, we clearly summarize and update the classification of asexual reproduction, influencing factors, molecular mechanisms, and the latest applications of apomictic genes in plant breeding, thus providing a reference for further research and the application of asexual reproduction in plants.

## 2. Classification of Asexual Reproduction

So far, reproduction through apomixis is limited to three common types: diplospory, apospory, and adventitious embryony [15]. The first two types involve the formation of unreduced embryo sacs and belong to gametophytic apomixis, whereas the latter is directly derived from ovule somatic cells and belongs to sporophytic apomixis [16] (Figure 1). However, parthenogenesis is also a common type of apomixis. The endosperm is needed to provide nutrients for seed formation and germination. In some asexual reproduction, the endosperm develops autonomously from the central cell, while the endosperm’s development in most apomixis requires pollination and fertilization (pseudogamy) of the central cell [16] (Figure 1). 

### 2.1. Diplospory

In diplospory, the embryo is derived not from the sporophytic cells of the ovary but from the egg cell of an unreduced embryo sac. Therefore, this is classified as a form of gametophytic apomixis together with the sporophyte. Furthermore, the unreduced embryo sac grows from the megaspore mother cell through restitutional meiosis or mitotic division, resulting in an unreduced embryo sac that is visually identical to a sexual embryo sac [16] (Figure 1f).

### 2.2. Apospory

Apospory refers to the direct development of an embryo from diploid cells around the embryo sac without meiosis. The most common type of apospory is one that develops from one of the two different types of ovule tissues: the nucellus or the inner nucellus [17]. In the process of aposporic reproduction, the development of an embryo is not derived from the megaspore mother cells in the ovary but from the ovule cells. Unreduced egg cells are produced through the unreduced embryo sac, ultimately forming offspring [16] (Figure 1e).

### 2.3. Adventitious Embryony

Adventitious embryony is one type of sporophytic apomixis, as embryos develop directly through the mitosis of ovule somatic cells [16]. This process does not involve the sexual embryo sac and is a type of asexual reproduction (Figure 1b).

### 2.4. Parthenogenesis

Gynogenesis and androgenesis both belong to the category of parthenogenesis, and they involve undergoing meiosis and mitosis but differ in terms of haploid embryo production.

#### 2.4.1. Gynogenesis

Gynogenesis refers to the process in which an unfertilized egg cell forms a seed and eventually spontaneously develops into an individual plant (Figure 1c). The genetic transmission of parthenogenesis requires only one gene or several closely related dominant genes. Gynogenesis, as the main reproductive mode of asexual reproduction, is an important approach for haploid breeding [18]. Gynogenesis can shorten the breeding period and allows the homozygous strain to be quickly obtained. However, the natural frequency of gynogenesis is low, so it cannot meet the needs of large-scale crop breeding. Egg division can be stimulated, and haploid plants can be obtained through distant pollen hybridization, induction with radiation or a chemical agent, interspecific hybridization, in vitro culture and in vivo induction, and the use of gynogenesis induction lines or other methods [19,20] (Figure 1c). In addition, scientists have found that gynogenesis-related genes expressed in egg cells can be inherited, and they play an independent role in apomeiosis and endosperm formation, thus inducing embryogenesis in polyploid, diploid, and haploid egg cells [18]. However, the mechanism of gynogenesis remains to be more deeply explored.

#### 2.4.2. Androgenesis

Unlike gynogenesis, androgenesis refers to the development of intact plants from embryos with only a male nuclear background [21] (Figure 1d). Compared with the production of double haploids through gynogenesis, the production of double haploids through androgenesis is more effective in many plants. Androgenesis was used to generate doubled-haploid (DH) plants from male gametophytes or their precursors via anther/microspore culture in vitro by undergoing a developmental transformation under a certain level of stress [22]. Usually, microspores in anthers develop into mature pollen grains through “microgametogenesis”. However, after certain treatments, the microspore development pathway could be transformed into an embryonic development stage, resulting in haploid or DH plants [23]. Double-haploid wheat was successfully obtained through androgenesis, and the following were the specific steps: the emasculation of wheat flower, the pollination of corn pollen, hormone treatment, embryo rescue, regeneration of haploid plants in tissue culture medium, and chromosome doubling [24]. Nevertheless, the process of obtaining DH plants through androgenesis still has defects such as genotype dependence, low frequency of embryogenesis, and regeneration; thus, further study and improvement are still needed (Figure 1d).

## 3. Progress of Apomixis-Related Genes

The conventional method of traditional asexual breeding involves genetically infiltrating sexually reproducing species and related species with apomictic traits, but this process is slow and difficult. Asexual plants are usually polyploids, and the characteristics of asexual reproduction itself also pose huge obstacles to the hybridization process. Moreover, the incompatibility of distant hybridization and gene redundancy causes great resistance to the creation of apomictic cultivars with good and beneficial characteristics. Therefore, in the last two decades, research has mainly focused on deciphering the underlying molecular mechanisms through genetic analysis combined with omics or other technologies to identify new and important regulatory genes or traits of apomictic crop breeding [25] (Table 1).

### 3.1. DYAD/SWITCH1 (SWI1)

To achieve the transition from sexual to asexual reproductive pathways in sexually reproducing crops, a feasible approach was proposed to manipulate the two key developmental processes of gamogenesis. Firstly, meiosis is converted into mitosis to produce maternal clonal gametes. Secondly, paternal-side genomic elimination occurs after the fertilization of female clonal gametes [73]. The first true non-fusion meiosis was observed in an *Arabidopsis thaliana dyad* mutant. *DYAD* encodes a nuclear entwining protein, which is necessary for entry into meiosis II; in *dyad* mutants, pollen development and male fertility are normal, while the megaspore mother cells fail to enter complete meiosis, as they are arrested at the end of meiosis I in the majority of ovules [32,33]. However, the fertile, undiminished female gametes produced by *dyad* mutants amounted to only 0.2%, which was far below the required frequency for crop-breeding applications. Thus, greater effort is needed to reach a higher haploid induction (HI) frequency (Table 1).

### 3.2. MiMe

Studies have shown that the *Ospair1*/*Osrec8* (*Ospair1*: Os03g01590, Pairing, *Osrec8*: Os05g50410, recombination protein) double mutant of *Arabidopsis* can produce diploid embryo sacs for the interruption of meiosis (Figure 2, Table 1). As for the interruption of meiosis, only one allelic division occurs, and then the embryo sacs stop their further development [26]. D’Erfurth et al. first identified one mutant of this gene and the other two key factors affecting meiosis [27]. The mutation of *ATSPO11-1* (*SPORULATION 11-1*) and *Atrec8*, which are involved in recombination and pairing, can regulate the separation of sister chromatids (Figure 2c,d, Table 1). Then, they also isolated and identified a new gene, *OSD1* (*OMISSION OF SECOND DIVISION*), in *Arabidopsis*, which is directly involved in the control of the entry into the second meiosis (Figure 2e). When combined with *OSD1*, the *Atspo11-1/Atrec8/Atosd1* triple mutant can produce functional diploid gametes with the same type as the parent, whose meiosis is completely replaced by mitosis; thus, this is called *Mitosis instead of Meiosis* (*MiMe*) [27] (Figure 2f). Compared with a *dyad* mutant, *MiMe* produced a higher frequency of unreduced gametes (100% male gametes and 85% female gametes), and the fertility of *MiMe* plants was not affected. Thereafter, *MiMe* and *dyad* mutants have been used as proofs of concept in an attempt to engineer apomixis in *Arabidopsis* [74] (Figure 3). The three genes that produce the *MiMe* genotype are highly conserved in plants, suggesting that asexual reproduction might be engineered in any plant species with this genotype, including crops [27] (Figure 2).

Subsequently, other studies have found that mutation in *SPO11-1* could be effectively replaced by the mutation of other recombinant initiation factors, such as *PRD1* (*PUTATIVE RECOMBINATION INITIATION DEFECT 1*), *PRD2*, or *PRD3*/*PAIR1*, thus also producing *MiMe* phenotypes. *AtSYN1*/*OsREC8* (*SYNCYTIN1*) encodes meiosis-specific cohesions, which could disrupt the single orientation of sister centromeres and lead to the early separation of sister chromatids during meiosis [75]. Finally, the transition from meiosis I to meiosis II requires *TAM* (*TARDY ASYNCHRONOUS MEIOSIS*, also known as *CYCA1*) and *OSD1*, and mutants of either of these two genes could lead to the production of diploid gametes [27,30]. By combining *TAM* with *Atspo11-1* and *Atrec8*, researchers also obtained plants that produced diploid gametes through mitosis-like division, and their genotypes were clones of their parents. The *TAM* allele showed a similar phenotype to that previously described in an *osd1* mutant [31]. *Ospair1* is an essential gene for the initiation of meiosis recombination in rice, and its mutant can reproduce the *spo11-1* phenotype in rice; this is the preferred gene for blocking homologous recombination in the construction of the rice *MiMe* phenotype [28]. *Os02g37850* is a homolog of the *AtOSD1* gene in rice, and the insertion of a mutant of this gene could also reproduce the *Atosd1* phenotype in rice, resulting in the omission of the second meiotic division. Therefore, the *Os02g37850* gene was named *OsOSD1*, and when combined with *Ospair1* and *Osrec8*, the *MiMe* phenotype was successfully established in rice [29]. The successful application of the *MiMe* technology in dicotyledonous *Arabidopsis* and monocotyledonous rice, which are far apart in terms of phylogeny, indicated that the *MiMe* technology should have potential revolutionary application value in crop improvement [29]. The diploid offspring produced by these combinations are genetically identical to the female parents because the fathers’ genomes are eliminated. Although the result of this artificial synthesis is the same as that of apomixis, it still requires hybridization. Nevertheless, this system demonstrates the feasibility of the clonal propagation of sexually reproductive species through seed engineering (Figure 2, Table 1).

### 3.3. CENH3

Centromere-specific histone H3 variant CENH3 (*CENTROMERIC HISTONE3*, also known as “CENP-A” in humans) is a ubiquitous protein in all eukaryotes [34,35,46]; it is essential for kinetochore nucleation and spindle attachment in mitosis and meiosis and has been effectively used for haploid induction [36]. CENH3 has a rapidly evolved N-terminal tail and a conserved C-terminal histone folding domain (HFD). Therefore, researchers found that the centromere construction of the *CENH3* variant appeared to be misassembled in hybridization with the wild type, thus resulting in haploids being inherited with only the wild-type parent; this can be used for haploid induction in plant breeding [37,38] (Figure 3). Ravi and Chan discovered that the null mutant of *Arabidopsis thaliana cenh3-1* had an embryonic lethal phenotype [39]. After that, many more studies were carried out to modify this gene. To our knowledge, all *CENH3*-based haploid induction applications that involve outcrossing wild-type plants with plants containing modified gene structures or *CENH3* gene expression variants have been classified into five categories to construct *CENH3* haploid induction (HI) lines (Figure 3; Table 1). 

#### 3.3.1. Tagged *CENH3* Variants

After the first finding of *cenh3-1*, Ravi and Chan manipulated the CENH3 protein. They found that the null mutants of *CENH3* complemented the altered version of CENH3, resulting in chromosome elimination in the mutant and eventually producing haploid plants when outcrossed with wild plants containing the conventional *CENH3* (Figure 3b) [39]. Two *CENH3* variants have been proposed: one is a green fluorescent protein (GFP) tagged at the N-terminal of *CENH3* (called *GFP-CENH3*), and the other involves replacing the N-terminal tail domain of *CENH3* with that from the *H3.3* variant (At1g13370), which is also tagged with *GFP* (called *GFP-tailswap*). Surprisingly, *GFP-tailswap* plants are almost male-sterile with a normal mitosis process, thus specifically indicating defective meiosis [39]. In addition, the *cenh3* mutant phenotype can also be restored by the hybrid combination of *cenh3-1* and *GFP-CENH3*. Moreover, the *GFP-tailswap* mutant could rescue the embryo’s lethal phenotype of *cenh3-1*, and the frequency of haploid induction was higher than that of *GFP-CENH3*. When a *GFP-tailswap* plant was crossed with wild-type *Arabidopsis* as the female parent, 25–45% of haploids containing only wild-type chromosomes could be produced. However, a lower haploid induction rate was obtained when this was used as the male parent, indicating that the elimination of the *CENH3* genome is more efficient when the HI line is used as the female parent. In addition, these tagged *cenh3* mutants can induce a centromere-mediated genome elimination, scaling down from a tetraploid to a diploid [39] (Figure 3; Table 1). 

#### 3.3.2. Untagged *CENH3* Knockouts

Recently, with the construction of genome-edited *TaCENH3α*-heteroallelic combinations using the CRISPR (Clustered Regularly Interspaced Short Palindromic Repeats)-Cas9 system, some researchers established a commercially operable parental HI line with a relatively high haploid induction rate (HIR) of ~7% in wheat (Figure 3c). Unlike that in *Arabidopsis*, heterozygous *TaCENH3α* wheat genotypes produced a higher HIR than homozygous combinations, thus paving the way for the development of *CENH3* HI technology in diverse crops using knockout *CENH3* lines without *GFP* tags [40] (Figure 3; Table 1). We speculated that the higher HIR came from the more disordered TaCENH3α in knockouts of heterozygous *TaCENH3α* than in homozygous combinations. In addition, TILLING (targeting induced local lesions in genomes) could also be used for the construction of untagged *CENH3* knockouts, thus speeding up the application of this technology in other crops [38] (Figure 3; Table 1).

#### 3.3.3. Orthologous *CENH3* Introgression Lines

It was demonstrated that the heterologous *CENH3* variants from *Lepidium oleraceum*, *Brassica rapa*, or other species could also be used to restore the embryo-lethal phenotype of the *cenh3-1* mutant in *Arabidopsis* (Figure 3d). When complementary assays were conducted with untagged orthologous *CENH3* sequences (N-terminal tail, HFD, or both) of *L. oleraceum*, the abnormal vegetative and reproductive phenotype in the *cenh3-1* mutant was restored. The obtained *CENH3*-complemented lines presented a genome elimination frequency of between 2% and 11% when outcrossed with wild-type *Arabidopsis*, providing another way to introduce the *CENH3* HI line into plants [37] (Figure 3; Table 1).

#### 3.3.4. In Vivo *CENH3*-RNAi Line

Apart from tagged and untagged *CENH3* variants and orthologous *CENH3* introgression lines, haploid plants can also be elicited by crossing with lines in which *CENH3* expression has been decreased in crops (Figure 3e). Recently, a *CENH3* transgenic RNAi line was used as a pollinator of wild-type parents to inhibit the native *CENH3*, thus simulating mock sexual reproduction and producing haploid progeny with a relatively high haploid progeny rate of ~8% in cotton. Therefore, the haploid bio-induction principle has been updated and provides a practical biotechnological guide for producing haploids in crops with the PTGS mechanism [41]. Recently, RNAi-mediated downregulation of *AcCENH3* also successfully mimicked an HI line in onion, thus significantly accelerating onion breeding [42] (Figure 3; Table 1).

#### 3.3.5. In Vivo Chemical Treatment Lines

Instead of traditional transgenic approaches, non-transgenic approaches can also be used to trigger haploid plants by outcrossing targeted plants with point mutant *CENH3* lines induced by chemical agents (Figure 3f). A report showed that point mutations resulting in single-amino-acid substitutions of the conserved HFD of *CENH3* induced uniparental genome elimination through EMS (ethylmethane sulfonate) treatment, resulting in seeds with haploidy or aneuploidy, with the retention of wild-type chromosomes [34,43]. Furthermore, single-amino-acid substitutions of P82S, G83E, A86V, A132T, L130F, and A136T in *AtCENH3* induced haploids with an HIR of 0.6–12 percent of progeny, depending on the mutation type. In addition, the selected A87V non-transgenic line was fully fertile in self-pollination and could produce postzygotic death and uniparental haploids when crossed with the wild type [43]. Similarly, lines treated with a *CENH3*-related inhibitor in vitro (CENH3 inhibitors: CYC (cycloheximide), ROS (roscovitine), MG115 (Z-leu-leu-norvalinal), and MG132 (Z-Leu-Leu-Leu-CHO)) displayed a decreased *CENH3* expression level comparable to that of transgenic RNAi lines in cotton; thus, it can also be used as a pollinator of wild-type parents to simulate RNAi HI for the production of haploid progenies through mock sexual reproduction [41] (Figure 3; Table 1). Therefore, these studies shed significant light on the introduction of a haploid induction mechanism into species that are reluctant to create transgenes through in vitro mutagenesis or inhibitor treatment with chemical agents.

At present, scientists are mainly focused on deciphering the underlying mechanism of *CENH3*-mediated haploid induction to advance the application of this technology in crop breeding. Some researchers suggested that the genomic disappearance occurred after the hybridization of the haploid induction line with the wild type, and the final generation of haploid plants might be the competitive outcomes between the modified and wild-type centromeres when developing hybrid embryos, resulting in the inactivation or loss of the inducer parent’s centromeres [34]. In addition, when producing haploids, *CENH3*-based HI lines can be applied to either female or male parents [39]. However, studies showed that a higher HIR could be generated when an HI line was used as a female parent [44]. Theoretically, due to the high conservation of *CENH3* in various eukaryotes, this technology holds significant promise for advancing the engineering of apomixis in crop breeding. In addition, an understanding of the potential mechanism of interspecific hybridization (a classical barley HI system involving crosses between *Hordeum vulgare* × *Hordeum bulbosum*), which can produce genome elimination, will be helpful for the explanation of the *CENH3* haploid induction mechanism [45].

### 3.4. MATL/PLA1/NLD

*MATL* is a pollen-specific phospholipase (pPLAII-α, *MTL/MATRILINEAL*) that is also known as *NOT LIKE DAD*, *NLD* [47] and *PHOSPHOLIPASE A1*, *PLA1* [48]; it was first found and identified in 2017. Compared with the non-inducible line B73, the maize induction line had a 4 bp insertion (CGAG) at the fourth exon of the inducible gene *MATL*, resulting in a 20-amino-acid frameshift mutation; thus, it could induce haploid maize [49]. Through the mutation editing of *MATL/PLA1*/*NLD*, a 6.7% haploid induction rate could be produced in maize [49]. Compared with *Arabidopsis thaliana*, which induces parthenogenesis through hybridization with *cenh3* variants, the advantage of *matl mutant* is that the selfing of *matl* mutants can directly result in parthenogenetic haploid offspring. At the same time, it was found that *MATL* is highly conservative, and its homologous gene exists in almost all monocotyledon but not in dicotyledon; thus, it can be extended to the genetic breeding of all other monocotyledonous plants. 

For example, the homologous gene *OsMATL* of *ZmMATL* was found in rice in 2018, and the *Osmatl* mutant produced 2–6% maternal haploid seeds when self-pollinated or cross-pollinated as the male parent [50]. In 2020, Professor Chen Shaojiang’s team at China Agricultural University and the Wheat Group carried out a gene function verification study based on the cloning of the key haploid induction gene *ZmPLA1* in corn, using allohexaploid common wheat as a model material [51]. The mutants of the wheat phospholipase gene *TaPLA1* were obtained through homologous gene cloning and gene editing. Experiments showed that the mutant could also produce about 2–3% haploid grains, while the growth and development of wheat and pollen viability were not affected in the *Tapla1* mutant [51]. Therefore, this is another good extension of the *ZmPLA1* system from maize to other monocotyledonous plants.

### 3.5. BABY BOOM 1 (BBM1)

*BABY BOOM* (*BBM*), a transcription factor of the AP2 (*APETALA2*)/ERF (*ETHYLENE RESPONSIVE ELEMENT BINDING FACTOR*) family, is a key regulatory factor of plant cell totipotency. The in vitro expression of *BBM* can induce embryogenesis without exogenous plant growth regulators or stress [52,53]. The *BBM* gene was originally found in *Brassica napus* (*B. napus*), where it has multiple functions in cell proliferation, plant growth, and the development of plant cells in the absence of exogenous growth regulators [52,54,55,56,57]. Among the genes involved in plant embryo development, the *BBM* gene plays an important role [52]. It is preferentially expressed when somatic cells are transformed into embryonic cells, and it can activate signal transduction pathways and induce differentiated somatic cells and somatic embryo formation [52]. It was found that the *BBM* gene is related to plant embryogenesis in many plant species. As a possible biomarker, the *BBM* gene is involved in a variety of functions, including cell proliferation, bud formation, somatic embryogenesis induction, development, promotion of apogamy, and stimulation transformation [52,53,57,58,59], but its molecular mechanisms are still unclear.

Studies have shown that the transgene of *PsASGR-BBML* (*PsASGR-Baby Boom Like*) can induce parthenogenesis in rice and maize, forming embryos without fertilization and eventually forming haploid plants [60,61]. Simultaneously, the CRISPR-Cas9 system was used to knock out *BBM1*, *BBM2*, and *BBM3*, and a termination or abortion phenotype in rice embryo development was observed. After reintroducing *BBM1*, it was found that normal embryos could be formed through parthenogenesis without fertilization. Then, the researchers combined *MiMe* with the ectopic expression of *BBM1* in egg cells and successfully established an asexual reproduction system in rice, which promoted the asexual reproduction of hybrid rice seeds [61].

*BABY BOOM 1* (*BBM1*) and *MATL*/*PLA1*/*NLD* are some of the genes reported to control parthenogenesis, and they have been reported to promote parthenogenesis in maize and rice [1,49,61]. *MATL* is highly conserved in monocotyledonous cereal crops, but homologous genes cannot be found in dicotyledonous plants [62]. Therefore, the *MATL* parthenogenetic induction system does not have a basis for widespread application in dicotyledons. However, *BBM* is highly conserved in both monocotyledons and dicotyledons, and it might be widely used in the asexual breeding of crops; thus, much more research should be conducted to elucidate the molecular mechanism of *BBM1* in the future, in addition to its applications. A recent study found that the binding of AtRKD5 to the 3′ end of *AtBBM* can inhibit the parthenogenetic potential mediated by *AtBBM*, which lays a solid foundation for the further exploration of the *BBM*-mediated parthenogenesis mechanism [63].

### 3.6. DMP

Shaojiang Chen’s team at China Agricultural University found that *ZmDMP* (*Zea mays DUF679 Domain Membrane Protein*) enhances and triggers haploid induction. Knockout of *ZmDMP* could cause a haploid induction rate of 0.1–0.3%, and in the presence of *mtl*/*zmpla1*/*nld*, it exhibited a greater ability to increase the HIR by 5–6 times. These findings provide an important way to study the molecular mechanism of haploid induction and improve the efficiency of maize DH breeding [64].

Recently, the same team found that the maize haploid-inducing gene *ZmDMP* is highly conserved in dicotyledonous plants through a comparison of gene sequence and successfully knocked out the *ZmDMP*-homologous gene *AtDMP* in *Arabidopsis* [65]. Finally, a haploid plant of *Arabidopsis* was obtained, thus extending the induction method for maize parthenogenesis haploids to the dicotyledonous plant *Arabidopsis* for the first time. In addition, a method for identifying *Arabidopsis* haploid seeds based on FAST-Red fluorescence was established with an accuracy of over 90% in this study [65]. This was the first successful application of *ZmDMP*-homologous genes in the study of haploid induction in dicotyledonous plants, thus laying the foundation for the establishment of a technical system for haploid seeds in dicotyledonous crops. Undoubtedly, *DMP* also has the potential to be widely used in the asexual breeding of crops after *BBM*. 

### 3.7. PAR

Researchers have discovered a dominant locus of parthenogenesis in dandelions (*Taraxacum officinale*) through the genetic analysis of apomixis [66]. Underwood et al. identified the *PAR* (*PARTHENOGENESIS*) gene, which encodes a zinc finger domain protein with an EAR (ethylene-responsive element-binding-factor-associated amphiphilic repression, DLNxxP) motif. They also detected the different expression patterns of *PAR* in apomictic and sexual dandelion tissues. This might be because of the insertion of a miniature inverted-repeat transposable element (MITE) transposon in its promoter, which was present in all dandelion apomicts tested. Moreover, the expression of dandelion *PAR* under the control of the egg-cell-specific promoter (*AtEC1*) indicated that *PAR* could induce the development of ectopic egg cell division and produce haploid-embryo-like structures in a relevant sexual crop—lettuce [67]. Furthermore, researchers also successfully produced haploid plants via the heterologous expression of the dandelion *PAR* gene in the egg cells of *Setaria italica*, opening the door to potential improvements in the application of DH technology in foxtail millet breeding programs [68]. Therefore, dandelion *PAR* has the potential for asexual plant breeding after the use of *BBM* and *DMP*.

### 3.8. RWP

Genes from a plant-specific family of transcription factors, *RWP* (*RKD*, *RWP-RK* domain-containing), have been shown to maintain egg-cell identity in *Arabidopsis* [69]. Then, researchers demonstrated that an MITE insertion in the promoter of the *CitRWP* (*Citrus grandis*) and *FhRWP* (*Fortunella hindsii*) genes resulted in polyembryony [70,71]. Apart from 10 sites with CHH methylation at the upstream end of the MITE sequences inserted into *FhRWP*, a transcription factor, *FhARID* (encoding an AT-rich interaction domain-containing protein), was found to target the MITE-containing *FhRWP* promoter of apomictic varieties, thus facilitating ovule gene expression and embryogenic induction [71].

Recently, researchers mapped the *MiRWP* (*Mangifera indica*) gene, which causes the polyembryony trait in mango and is orthologous to the *CitRWP* gene, which causes polyembryony in citrus. Based on these results, they speculated that nucellar embryogenesis was caused by promoter insertion events, which transpired independently in citrus and mango. The findings imply that polyembryony evolved convergently in the two species. However, much more research is needed to determine whether these genes (citrus and mango) can be used as a tool for the clonal production of other crops in different biological systems [72].

## 4. Main Factors Affecting Apomixis 

Apomixis is influenced by genetic and ploidy factors, as well as the regulation of apomixis-related genes. In addition, it is influenced by differential gene expression, epigenetics, hormones, plant genomic evolution, and so on.

### 4.1. Differential Gene Expression

One report showed that the sexual and apomixis pathways in *Hieracium* shared the same gene expression profile and molecular regulation characteristics, thus indicating that they have closely interrelated developmental pathways [76]. It seems that the form of apomixis (a combination of apomixis with autonomous embryo and endosperm development) differs from gamogenesis in limited ways. Nevertheless, it is similar to gamogenesis except for two specific turning points: meiosis and fertilization. Some scientists suggest that apomixis is the relaxation of gamogenesis in both space and time, which leads to a change in cell fate and the absence of key steps in sexual development [2]. Therefore, studies and discoveries of specific or differentially expressed genes in embryo and embryo sac formation may be breakthroughs in the research process for apomixis [77].

Although there are few studies on the molecular regulation mechanism of apomixis, some achievements have also been established. The expression pattern of some genes plays an important role in apomixis (Table 2). Polycomb Group (PcG) of the multicomb family is one of the regulatory cores in the apomictic process, and its mechanism is based on two main types: Polycomb Repressive Complex 1 (PRC1) and PRC2 [78]. Changes in the expression level of the *PRC2* gene may result in abnormal seed development in *Arabidopsis* and rice [79,80]. Other important members of the Polycomb taxon, including the *FIS* gene (*FIS*, *FERTILIZATION INDEPENDENT SEED*) and *FIE* gene (*FIE*, *FERTILIZATION-INDEPENDENT ENDOSPERM*), also play a role in apomixis. The *FIS* gene plays an important role in plant growth and development, and there are significant differences in the expression level of *FIS* between sexual and asexual apple varieties. It is speculated that the *FIS* gene may participate in the reproductive regulation of apomictic varieties of apple [81]. In *Arabidopsis fis* mutants, embryo degeneration and overdevelopment of the endosperm were observed [82]. Silencing of the *FIE* gene leads to abnormal phenotypes during reproductive development in tomatoes, such as increasing numbers of sepals and petals, the fusion of ovules and pistils, and parthenocarpy [83]. In addition, *AGAMOUS-like 62* (*agl62*) can combine with *FIS* to form a complex, which mediates the occurrence of 65 kinds of endosperm cellulose [80].

Moreover, mutation of the *ARGONAUTE* genes in sexual plants can change the number of certain cells and has the ability to initiate embryo sac development [85,86]. The *argonaute9* (*ago 9*) mutant in *Arabidopsis* appeared to form multiple cells within a single ovule, with the potential to undergo mitosis and to evoke an embryo sac, thus providing a primitive cell source for the formation of multiple embryos [85]. The maize *argonaute104* (*ago104*) mutant also displayed an asexual phenotype. Additionally, the phenotype of *AGO104* is similar to that of spores, with one megaspore mother cell per ovule undergoing mitosis to produce diploid gametes rather than meiosis [86]. The accumulation of the AGO104 protein in somatic cells around megaspore mother cells suggests that this mobile signal is responsible for inhibiting the somatic cell fate of germ cells and that the *AGO* gene is also related to germ cells in rice [28].

The differences in meiotic chromosome organization between male and female meiocytes in an *Arabidopsis dyad* mutant indicate that the *DYAD* gene could regulate meiosis [87]. In addition, *ORIGIN RECOGNITION COMPLEX* (*ORC*) is a multiprotein complex that controls DNA replication and cell differentiation during fusion-free processes [88]. *GID1* (*GIBBERELLIN-INSENSITIVE DWARF1*) may be involved in single megaspore mother cell differentiation during ovule development and plays a role in sexual and apomictic reproduction [62]. The *MSP1* (*MULTIPLE SPOROCYTE1*) gene encodes a Leu-rich repeat receptor-like protein kinase and controls the number of megaspore mother cells in rice. Instead of being expressed in archesporial cells and megasporocytes, the *MSP1* gene is expressed in somatic cells around archesporial cells, and it inhibits the sporulation of somatic cells around archesporial cells and the formation of the anther wall, thus making somatic cells unable to differentiate into germ cells. In contrast, the *msp1* mutant produces a large number of male and female spore cells, resulting in disrupted formation of the anther wall and loss of the tapetum [89].

*PpSERK* (*SOMATIC EMBRYOGENESIS RECEPTOR-LIKE KINASE*) and *APOSTART* are considered to be involved in intercellular signal transduction and hormone transport [7]. It was found that the activation of *PpSERK* in nucellus cells of the apomictic genotype is a switch to start embryo sac development, and it can redirect the signal gene products to the atypical intercellular region. The *SERK*-mediated signaling pathway may interact with the *APOSTART*-controlled growth hormone pathway. *APOSTART* contains a lipid-binding START domain that is thought to play a role in meiosis and may also be associated with programmed cell death and nonfunctional megaspore degeneration. One of the two segregating copies (*APOSTART1*) associated with apomixis is overexpressed in sexual germ lines. The phenotype of the *apostart1/apostart2* double mutant suggests a role for *APOSTART1* in embryo and seed development [77]. Subsequently, the crucial regulator that transforms a “normal” nucellar cell into an initial apospory is called *APOSTART_6* [90]. 

In addition to the genes mentioned above, many other genes are also involved in the apomixis, such as *WUSCHEL* (*WUS*) [91], *HpMEE29-like* [92], *HpARI7* [93,94], *PnTgs1*-like [95], and so on (Table 2).

### 4.2. Epigenetics

The epigenetic regulation of apomixis is an attractive (and probably true) hypothesis because it can probably explain why facultative apomixis occurs and why apomixis can return to gamogenesis. Evolutionarily speaking, epigenetic changes may occur after polyploidy, activating apomictic expression and inhibiting gamogenesis. In this case, all angiosperms might have the “potentiality” for apomixis [96]. An increasing number of studies have shown that DNA methylation plays an important role in apomixis and heterosis. Genetic and epigenetic regulations are important determinants of plant evolution, adaptation, and plasticity [97]. DNA methylation and other chromatin modifications, such as small RNA regulation, nucleosome remodeling [98], histone covalent modification, gene silencing, and RNA editing, co-regulate or separately influence some important biological processes. Among these, DNA methylation is one of the most clearly understood epigenetic phenomena [99]. A report showed that methylation levels in sexual plants are higher than those in apomictic plants. It was found that more than 95% of cytosines were methylated in the analyzed fragments of sexual plants, while only 35% of cytosines were methylated in apomictic plants [100]. However, the causal link between epigenetic mechanism(s) and apomixis has not yet been established.

The protein structure of *DNMTs* (DNA cytosine-5’-methyltransferases) is common in *Boechera* species that engage in both asexual and sexual reproduction. Three methylation-related genes—*MET1* (*METHYLTRANSFERASE1*, methyltransferase), *CMT3* (*CHROMOMETHYLASE3*, chromatin methyltransferase3), and *DRM2* (*DOMAINS REARRANGED METHYLTRANSFERASE2*)—are conserved in green algae, monocotyledons, and dicotyledons, but the expression levels of *DNMT* genes are different in apomictic and gamogenetic species. For example, *DRM2* was up-regulated in asexual species after fertilization [101] (Table 2). In maize (*Zea mays*), *DMT102* (*DNA methyltransferase102*) and *DMT103* (*DNA methyltransferase103*) were down-regulated in ovules, resulting in the formation of undiminished gametes and multiple embryo sacs. Thus, an active methylation pathway during maize reproduction is required for gametophyte development and plays a key role in the differentiation of sexual and apomictic reproduction [84]. It was also found that *DMT102* and *DMT103* genes were mainly expressed in germ cells and their surrounding restricted regions [84]. Furthermore, the methylation levels of genome segments controlling apomixis were very high in *Paspalum notatum* and *Paspalum simplex*. Artificial demethylation had little effect on apospory, but the repression factors of parthenogenesis may be inactivated by DNA methylation [102]. 

In the DNA methylation process of *Boechera*, compared with sexual species, meiotic homologous genes such as *ASYNAPTIC 1* (*ASY1*), *MULTIPOLAR SPINDLE 1* (*MPS1*), and *NAC019* were down-regulated in apomictic individuals [103]. Furthermore, citrus nucellar polyembryony (NPE) is a type of sporophytic apomixis in which adventitious embryogenesis from somatic nucellar cells forms asexual embryos in the seed. DNA hyper-methylation may activate *CitRWP* transcription, boosting *C2H2* expression and ROS accumulation, triggering epigenetic regulation, and controlling cell fate transition and nucellar embryo initial (NEI) cell identity in apomictic cells, according to the working model for citrus NPE initiation [104].

The role of the RNA-dependent DNA methylation pathway (RdDM) in female reproduction may also be a key regulator of apomixis and sexual differentiation [105], and it can down-regulate the expression of related genes [106]. In *B. napus*, the existence, deletion, and distribution of *BnMET1a-like* DNA methyltransferase transcription are related to methylation. However, methylated genes involved in pollen reprogramming can turn pollen growth and development into pollen embryogenesis, and the DNA methylation and *MET1a* expression patterns in such embryos are similar to those in zygote embryos [107]. Moreover, a well-studied class of small regulatory RNAs found in many eukaryotes seem to control the function of gametes and plant fertilization by modifying the expression of certain genes through translational inhibition, post-transcriptional gene silencing, and heterochromatin alteration. A silencing assay of *AGO9*-dependent sRNA showed that *AGO9*-dependent sRNA plays a key role in the cell fate of ovules and epigenetic reprogramming in related cells of plant gametes [85] (Table 2). Combining the available small RNA profile data with previously published transcriptome data from the flowers of sexual and apomictic *P. notatum* plants, some researchers speculated that the auxin pathway regulated by sRNA plays a critical role in apomixis promotion. Future functional characterization of the molecular elements driving the transition from sexuality to apomixis could be carried out based on their results [108]. 

The epigenetic regulation of apomixis is an attractive theory that deserves more attention, as it potentially accounts for the facultative nature of apomixis. The epigenetic regulation of some important genes or small RNAs might be a key factor that can speed up our understanding of the apomictic mechanism in different species. Therefore, more studies should be conducted on the epigenetics of DNA methylation, RdDM, and small RNA expression.

### 4.3. Hormones

Recently, studies showed that the differentiation of aposporous initial cells (AICs) may also be related to the disordered RdDM pathway, chromatin remodeling proteins, and hormone homeostasis [109]. Several genes involved in cytokinin biosynthesis were down-regulated in aposporous pistils (Table 2). Other experimental data reported that the regulation of the transcription of genes involved in auxin homeostasis, whether through intracellular transport or the hydrolysis of IAA conjugates, is associated with aposporous gametophyte development [110], suggesting that phytohormones play an important role in apomixis.

In another study, *Pin-formed 8* (*PIN8*) regulated intracellular auxin homeostasis and affected auxin-regulated gene transcription by blocking auxin’s entry into the nucleus or altering the biological activity of the nucleus. They demonstrated that the regulation of gene expression involved in auxin homeostasis through intracellular transport or hydrolysis of IAA binding and the development of spore-free gametophytes coincided [110]. Similarly, the growth hormone exclusion vector gene *PIN-FORMED 1* (*PIN1*) is required for the development of blastocysts in *Arabidopsis* [111], and the inhibition of the polar growth of hormone transport affects the timing, location, and frequency of aposporous initiation cell formation [5]. In addition, the absence of functional macrospore-specific markers in the ovules of *arabidopsis histidine kinases 2* (*ahk2*), *ahk3*, and *ahk4/cre1/wol* triple mutants suggested that disruption of cytokinin signaling blocks functional macrospore formation [112]. Thus, the changes in auxin or cytokinin signal transduction may play an important role in determining the identity of the embryo sac of the aposporic initiation cells in aposporous plants. 

It has been proposed that *PpSERK* activation in nucellar cells of apomictic genotypes is the switch that guides embryo sac development, and it can redirect the signaling gene products to compartments other than their typical ones. In addition, the SERK-mediated signaling pathway might interact with the auxin/hormonal pathway controlled by *APOSTART* at some point, though it is not an integral part of it [7] (Table 2). Assays on the accumulation or flow of auxin might provide sporophytic information on the apomictic reproduction process in *Hieracium*. Using N-1-naphthylphthalamic acid (NPA) to inhibit the polar transport of auxin in flowers from apomictic Hieracium could induce changes in apomictic reproductive development; this supports the opinion that auxin transport alteration leads to the acellular autonomic deregulation of apomictic reproduction [5]. 

In addition to polyamine and spermidine metabolism, other important hormonal pathways were observed in apomictic initial cells. Unlike in mature gametophytes, up-regulation of cytokinin degradation was detected in an apomictic germline specification [113]. Another study showed that cytokinin biosynthesis and key components of the reaction regulator are expressed during ovule development [114]. In the culture of unfertilized ovules, different steroid hormones induced the development of central cells into an unfertilized endosperm, confirming the ability of steroid hormones to facilitate the transition from sexual endosperm development to apomictic endosperm development in *Arabidopsis* (Col-0) [115]. For instance, some researchers demonstrated that dimethyl sulfoxide (DMSO) could induce apomixis in cassava [116]. One study showed that in vitro treatment with 1%, 1.5%, and 2% (V/V) dimethyl sulfoxide (DMSO) on female buds could also trigger apomictic reproduction in cassava, and 1.5% DMSO treatment gave the most effective apomixis induction rate [116].

All of these studies indicated that hormone regulation plays an important role in apomixis. Therefore, finely deciphering hormone regulatory pathways can also help in understanding the apomictic mechanisms in plants.

### 4.4. Plant Genomic Evolution

The repetitive content of the plant genome (repeatome) typically represents its largest fraction and is frequently correlated with its size [117]. Transposable elements (TEs) are DNA fragments that can move within the genome and have high mutagenic potential for the host genome [118]. TEs are the main component of the repeatome, and due to their fast-evolving nature, they are an important driver in genome diversification [117]. A study found that *AGO9* could preferentially interact with 24 nucleotide (nt) sRNAs derived from TEs, and its activity was necessary for silencing TE in female gametes and their helper cells [85]. In addition, another study found that the dominant *PAR* allele has an MITE transposon insertion in the promoter. A promoter containing MITEs can invoke a homologous gene from sexual lettuce to supplement dandelion *loss of parthenogenesis* mutants. A similar MITE is also present in the promoter of the *PAR* gene in apomictic forms of hawkweed, indicating parallel evolution [67]. The main citrus varieties with the polyembryonic allele produced polyembryonic seeds. Therefore, inserting MITE into the upstream region of dominant *CitRKD1* might be involved in regulating the transcription of *CitRKD1*. Moreover, researchers have found heterozygous structural variants (SVs) in the *FhRWP* and *CitRWP* promoters of apomictic citrus and *Fortunella* due to the insertion of two or three MITEs. The transcription factor *FhARID*, which encodes an AT-rich interaction domain-containing protein, binds to the MITEs in the promoter of apomictic varieties, promoting the induction of nucellar embryogenesis. This study provides evolutionary genomic variation and molecular insights for apomictic reproduction in citrus [71].

## 5. Application of Apomixis in Sexually Reproducing Plants

In recent years, the introduction of apomixis into sexually reproducing plants has become the trend of its development. Some researchers demonstrated that apomixis allows the transgenerational fixation of phenotypes in *Hieracium pilosella* hybrids across generations [119]. This provides a fundamental reference for the use of apomixis in plant breeding and seed production studies to fix complex (or hybrid) quantitative phenotypes across generations.

### 5.1. Haploid Induction Line

A HI line can induce haploid production with higher frequency after hybridization between parental and parental receptor plants. This has become the main method of haploid induction in maize, and it involves three steps: (1) using the inducer genotype to pollinate the source germplasm to produce the parent haploid seed; (2) distinguishing haploid seeds from normal diploid crossing seeds; (3) doubling the chromosomes of haploid plants [120]. In the 20th century, the construction of inbred maize lines was almost dependent on self-crossing, which required 6 to 10 generations to reach the desired level of homozygosity [14]. Chase first proposed the double haploid (DH) technique, which could result in pure lines within two generations, thus greatly reducing the time needed for inbreeding [121].

In recent decades, DH techniques based on in vitro HI have emerged as some of the most important tools in maize breeding, and they have replaced conventional strain development through cyclic self-pollination [122]. Stock 6 is the most widely used maize haploid inducer at present. When the inducer line derived from Stock 6 was used as the paternal parent, haploid individuals in which only the genome information from a female parent was retained could be formed [123]. Maize haploid matrix lines are often obtained through the pollination of an HI system (such as Stock 6, with a haploid induction rate of 1–2%) [124], or another inducer is derived from the Stock 6 inducer (such as the CAU5 inducer). The induction rate of maize breeding programs in many parts of the world is as high as approximately 12% [125]. By incorporating the PEM (Purple Embryo Marker) marker gene system, BP1A1A2C1C2, the *R-nj* method can be used to efficiently identify maize haploids [126].

Outside of maize, a new in vivo route for producing maternal double haploid of *B. napus* came into being: the pollen donor, an allooctoploid rapeseed, acts as an allo-octoploid DH inducer in *B. napus*, in which there might be a novel mechanism for haploid induction. Maternal double-haploid *B. napus* was yielded by crossing rapeseed in the maternal line with allogeneic octoploid rapeseed and had a relatively high induction rate [127]. In maize and *Arabidopsis*, the early haploid induction rates were 1.2–16% [128] and 25–45% [39]. However, there was no significant difference in the induction ratios of the two *Brassica* octoploids Y3560 and Y3380, and their induction ratio for cytoplasmic male sterile *B. napus* lines was around 90%, which was much higher than that (nearly 40%) for genic male sterile *B. napus* lines. This indicates that the induction rate depends on the genotype of female *B. napus* (genotype-dependent); compared with other haploid induction lines, the *Brassica* octoploids Y3560 and Y3380 may have more practical and economic value [127]. This research provides new insights into a single-step method of generating homozygous lines in vivo, thus promoting the understanding of cruciferous plant breeding programs and genetic research [127]. 

Another study showed that DH inducibility is attributed to key functional gene regulation from the parents (P3-2) of Y3560 and Y3380, which significantly increases the induction efficiency in comparison with that of ploidy and can be inherited through generation screening [129]. Soon after that, the same team also found that some plants of the induced offspring had a small amount of paternal gene infiltration, which was similar to that in the maize haploid inducer line, and the induction differences were affected by the interaction with the maternal genotype and cytoplasmic type [130]. Recently, researchers speculated and demonstrated that the chromosomes of the induced line (parental line) were eliminated during embryonic development, while the maternal haploid chromosomes were synchronously doubled in the embryo. All of these findings helped for the future localization or cloning of functional genes involved in the *Brassica* allo-octoploid DH inducer [131].

The haploid induction abilities in different DH-inducing plants might be controlled by a specific quantitative trait locus (QTL). To date, eight paternal [120,132] and two maternal QTLs [133] were found in maize haploid inducers. Haploid induction ability in maize is mainly controlled by two key quantitative trait loci, quantitative haploid induction rate 1 (*qhir1*, also known as parthenogenesis inducer1, *ggi1*) and *qhir8*, which account for approximately 66% and 20% of the genetic variance in the HIR, respectively [1,120,134]. The *ggi1*/*qhir1* locus is considered the controlling site of haploid induction ability in maize, and it is related to the segregation distortion caused by poor pollen transmission by the inducer [120,134]. The *ggi1*/*qhir1* locus has been shown to induce genome elimination and has been widely used in maize haploid breeding. In chromosome doubling breeding, potential genes at the *ggi1*/*qhir1* site would open up exciting new approaches to asexual reproduction [132,134] and may induce the parthenogenetic development of embryos, etc. [1]. Mutations in the *MTL*/*ZmPLA1*/*NLD* gene at the *qhir1* site in Stock 6 can produce a HIR about 2% [120] (Table 1). The second important QTL, *qhir8* (located on chromosome 9), explains 20% of genotypic variation and is significant in the improvement of the haploid induction ability of *ggi1*/*qhir1* [135]. The mutation of a non-Stock 6 source gene *ZmDMP* in *qhir8* could also enhance and trigger haploid induction. *Zmdmp* knockout mutants generate 0.1–0.3% HIR, but in the presence of *mtl*/*Zmpla1*/*nld*, they could increase the HIR by 5–6 times [64] (Table 1).

### 5.2. Applications of Apomictic Genes 

As we have discovered so many apomixis-related genes, appropriate modifications and suitable applications would facilitate plant improvement. For example, by combining studies of gene-editing methods and the discovery of the haploid-inducing genes *MATL/PLA1/NLD* and *ZmDMP* in maize, haploid induction systems can be obtained efficiently and quickly in monocotyledonous and dicotyledonous plants, thus accelerating plant-breeding progress [44,50,51,65]. Recently, with the application of the CRISPR/Cas9 gene editing technology, a Chinese group successfully obtained *MiMe*-type Chunyou 84 rice by simultaneously knocking out three endogenous meiosis-related genes (*PAIR1*, *REC8*, and *OSD1*) and then knocking out *MATL*, which encodes pollen-specific phospholipase in *MiMe*-type rice. The obtained quadruple-mutant plants produced a large number of tetraploid and diploid seeds, and the acquisition rate of diploid cloned seeds was 6.2% [44]. Although there is a significant decrease in fertility, this is a successful combination of *MiMe* with rice apomixis without introducing hybrid pollination, and it is a breakthrough in the application of *MiMe* [44]. This breakthrough successfully introduced apomictic reproduction characteristics into Chunyou 84 and resulted in tetraploid clone seeds, enhancing the fixation of hybrid rice to maintain its heterozygosity. It was found that there were no differences in the growth or morphology between three *MiMe* mutants and wild-type CY84, and the editing of these three genes did not significantly affect the fecundity of the hybrid [44]. Similarly, in the same year, other researchers mutated *OsSPO11-1*, *OsREC8*, *OsOSD1*, and *OsMATL* with the CRISPR/Cas9 system to introduce apomixis into rice (*Oryza sativa*). Because of its ability to generate apomictic offspring, they named this quadruple mutant *AOP* (*Apomictic Offspring Producer*) [136].

At the same time, other researchers also found that the combination of the ectopic expression of AP2 family transcription factor *BBM1* in egg cells and the *MiMe* genotype induced parthenogenesis in rice and achieved asexual reproduction of rice seeds [61]. Studies have documented that the apomixis-related gene *PSASGR-BBML* can induce parthenogenesis when expressed in egg cells with reduced chromosomes in various cereal crops (pearl millet, maize, rice, and sorghum). The rice *PsASGR-BBML* gene is closely related to the rice homolog *BBM1* gene, whereas the rice *BBM1* protein is expressed in pollen, is delivered to the oocyte at fertilization, and thus plays a role in the initiation of post-fertilization embryogenesis [137]. This indicates that the rice *PsASGR-BBML* gene and the rice homolog *BBM1* gene are different orthologues with the same function. Therefore, further research on related genes may enhance the application of apomixis in crop breeding.

*MiMe* and *dyad* mutants do not copy parthenogenesis. However, when crossed with *Arabidopsis cenh3 variants*, a target plant with cloned diploid seeds was successfully obtained, achieving the successful engineering of apomixis in *Arabidopsis thaliana* [39]. *CENH3*^−/−^ and *CENH3*: RNAi maize lines can be complemented with *AcGREEN-tailswap-CENH3* or *AcGREEN-CENH3* transgenes. When crossing these *Zmcenh3* mutants with wild-type plants, the highest HIR was 3.6%, and the average HIR was about 1%. Though the frequency of HI in maize is relatively low when compared with the HI of *GFP-tailswap* used in *Arabidopsis* by Ravi, it is of great importance for the advancement of apomixis engineering in plant breeding [36,39]. In addition, the combination of *MiMe* and the *CENH3* chromosome elimination technique also produces artificially cloned haploid offspring in rice [29].

Currently, *MiMe* plants have been successfully obtained in *Arabidopsis*, rice, and other species [29,36,39,44,61,136]. However, the expansion of this to more species is an important issue for the applications of *MiMe*. Another difficulty is expanding apomictic characteristics to other plants by using homologous genes related to apomixis in the model plant, *Arabidopsis thaliana*, as there might be different meiotic mechanisms across various species [138]. According to recent research, hybrid rice that expresses dandelion *PAR* specifically in its egg cells in conjunction with *MiMe* can effectively initiate highly fertile synthetic apomixis, significantly facilitating the successful clonal propagation of hybrids [139]. In addition, multiple copies of genes controlling the cell division cycle may exist in plants, such as *osd1*, which is a single gene in barley and oilseed rape but exhibits tandem repeats in maize, sorghum, and *Setaria viridis*. Therefore, RNAi or other genome-editing techniques such as CRISPR/Cas used in the construction of *MiMe* phenotypes also make the application of studies of apomixis more complex. *MiMe* technology produces apomeiosis gametes, and parthenogenesis enables the production of asexual gametes to initiate embryonic development. However, the initiation of endosperm development is also required to ensure seed formation when combining *MiMe* with other apomixis-related genes.

In addition to genes that have potential effects on the key steps of apomixis and genes related to the development of apomixis embryos, concurrently, a fascinating diversity of proteins, small RNAs, and long non-coding RNAs have been found to functionally characterize apomixis-related genes and regulatory sequences. These RNAs are most likely parts of the regulatory cascade of apomixis; therefore, further research is still needed to determine their overall function and connectivity [109]. In summary, these studies further confirmed the feasibility of asexual reproduction in sexual plants, laying the foundation and providing a new dimension for the application of apomixis in the genetic breeding of plants/crops.

### 5.3. HI-Edit

At present, instead of the traditional pedigree breeding method, the hybridization-induced haploid breeding method has become the mainstream breeding strategy, resulting in the rapid creation of inbred DH lines in a short time (within two generations) regardless of the genetic background. Apomictic genes can be utilized for haploid induction through genetic editing, overexpression, and other transgenic strategies, in addition to being applied through chemical treatments to obtain homozygous plants or preserve hybrid vigor [29,36,39,44,61,136]. However, the development of genome-editing technology, represented by CRISPR and CRISPR/Cas9 gene editing (Edit), has become an indispensable method for genome modification and gene function studies, thus further helping accelerate the progress of apomixis in plant breeding [140,141]. 

In 2019, a report from Syngenta Plant Protection in the United States proposed a new strategy called HI-Edit, which combined haploid induction (HI) with CRISPR/Cas9 gene-editing (Edit) technology to directly achieve haploid induction with improved crop breeding through genomic editing of specific agronomic loci at the same time [140]. This efficient and fast target-gene-editing breeding system is a new milestone in the history of crop breeding. This system was divided into three steps: first, the researchers completed the editing of specific sites in commercial corn germplasm with the paternal parent *matl* using the CRISPR/Cas9 gene-editing strategy. Second, through the gene editing of several genes related to the number and weight of corn kernels (*VLHP1*, *VLHP2*, *ZmGW2-1*, *ZmGW2-2*), hybridization experiments were conducted to verify that the HI-Edit strategy can be used for the improvement of breeds of corn crops. Then, through related experiments in *Arabidopsis thaliana*, an improved *CENH3* HI-Edit system that can be used in dicotyledonous plants was developed. Finally, wheat genes were successfully edited with a combined strategy of the HI-Edit system and inter-genus hybridization [140]. Nearly at the same time, Chinese scientists also came up with another new breeding strategy called IMGE (Haploid Inducer-Mediated Genome Editing), which can create double-haploid maize lines with improved non-transgenic traits within two generations, thus also enabling gene-editing abilities in crop breeding. To verify whether an HI line carrying a CRISPR/Cas9 cassette can be used to create new edited haploid plants or not, a *CAU5* HI line carrying the CRISPR/Cas9 cassette for *ZmLG1* and *UB2* was used to conduct experiments, and they successfully generated genome-edited haploids in maize [142]. These studies provide new technologies and dimensions for research on haploid induction breeding and transgene-free breeding. 

Thus, these results indicate that the HI-Edit can not only rapidly introduce *MiMe* phenotypes or haploid induction types using the modification of apomixis-related genes in hybrid plants, but also genes of interest can also be edited by CRISPR/Cas9 gene-editing simultaneously, thus greatly promoting the application progress of personalized breeding for apomixis. HI-Edit especially breaks through the limitations of species and genotypes in existing gene-editing systems and directly modifies the genomes of commercial crop varieties, with broad application prospects.

### 5.4. Other Applications of Natural Apomixis

Most species of citrus have apomictic abilities and can produce multiple embryos. Therefore, it is of great significance to strengthen related research on this special biological phenomenon in citrus, and it is promising that its apomictic reproduction regime or mechanism can be used to fix the heterosis of important grain and oil crops. Some scholars have proposed the dual-gene regulation hypothesis of apomixis in citrus plants, which can explain most of the existing hybrid separation data in citrus [143]. In addition, the natural occurrence frequency of apomictic genes is low, and gene transfer can be carried out by crossing cultivated cassava with wild species to increase the apomictic rate in cultivated cassava. Apomixis in cassava is controlled by more than one recessive gene, and these recessive genes act with additive effects. Aneuploidy in cassava is related to apomixis and can provide more recessive genes, thus indicating that polyploidization can increase the rate of apomixis. From an evolutionary point of view, combining polyploidy and apomixis contributes to the production of new species [144]. 

Similarly to the initiation of embryonic development through parthenogenesis, there are many special mechanisms of endosperm formation in the natural world. For instance, it was found that the endosperm in *Hieracium panicalatum* and *Eulaliopsis binata* (Poaceae) was generated through autonomous division without fertilization [145]. If this autonomous endosperm generation mechanism could be combined with current asexual division and parthenogenesis technologies, a thorough revolution in the asexual reproduction of sexual species could be achieved, where apomictic seeds could be produced without fertilization. Furthermore, studies of the regulatory network between the endosperm development mechanism and genes that can produce the *MiMe* type will also provide new promise for the application and study of apomixis.

## 6. Prospects and Conclusions

Conventional apomixis is achieved through in vitro tissue culture, cuttings of a few varieties, or the hybridization of targeted variants with natural haploid induction lines. With the development of biotechnology and the advancement of research, researchers have increasingly clarified the nature of apomixis and the related molecular mechanisms. Apomixis can regulate plant reproductive processes to form large clonal populations with consistent genotypes and fix hybrid vigor through successive generations of self-crossing seeds to maintain genetic purity from one generation to another [146]. Although the great potential of apomixis in heterosis fixation and crop breeding has been shown, no significant breakthroughs have been made in the delivery of apomictic traits into major crops due to defects such as high seed abortion in apomictic hybrids. Therefore, establishing a stable and universally applicable system of apomixis for plant breeding still requires a complete understanding of the genetic mechanisms in natural apomicts. Future research on apomixis will mostly focus on multiple combinations of modern genetics and epigenetic regulation mechanisms that control apomixis [25], the exploration of mechanisms of apomixis in apomictic populations of different origins, the identification of apomicts, the characterization of apomixis-related genes, and some other important issues. Deciphering more apomixis-related genes and mimicking the apomictic phenotype in sexual crops will strongly boost the penetrance of apomictic crops [16]. Moreover, the emergence of laser-capture microdissection, targeted gene editing, and other efficient plant transformation systems will largely promote our understanding of genes and networks that are related to apomixis and will ultimately contribute to the application of clonal reproduction in sexual crops [16].

Compared with conventional genetic improvement, the application of apomixis in plant breeding would significantly increase the varieties with superior traits, take advantage of hybrids, and reduce the input cost with a high yield, making their quality and yield more uniform (Figure 4). Conventional genetic breeding generally takes 6–8 generations before stable line varieties for production are obtained (Figure 4a). In contrast, apomictic breeding requires approximately two generations, significantly shortening the breeding period with the target characteristic. Apomictic breeding can reduce the ploidy of plants, facilitate easy plant breeding (Figure 4b), play a significant role in maintaining hybrid vigor (Figure 4c), and even fix new genomically edited agronomic loci (Figure 4d) in backgrounds, thus greatly promoting the development of plant breeding. 

In conclusion, we have explored the classification of asexual reproduction, apomixis-related genes (*DYAD*, *MiMe*, *CENH3*, *MATL/PLA1/NLD*, *BBM1*, *DMP*, *PAR*, *RWP*, etc.), the main factors influencing apomixis (differentially expressed genes, epigenetic regulation, hormones, plant genomic evolution, etc.), and the applications of apomixis in order to provide some references for future research on the apomictic breeding of plants.

## Figures and Tables

**Figure 1 ijms-25-11378-f001:**
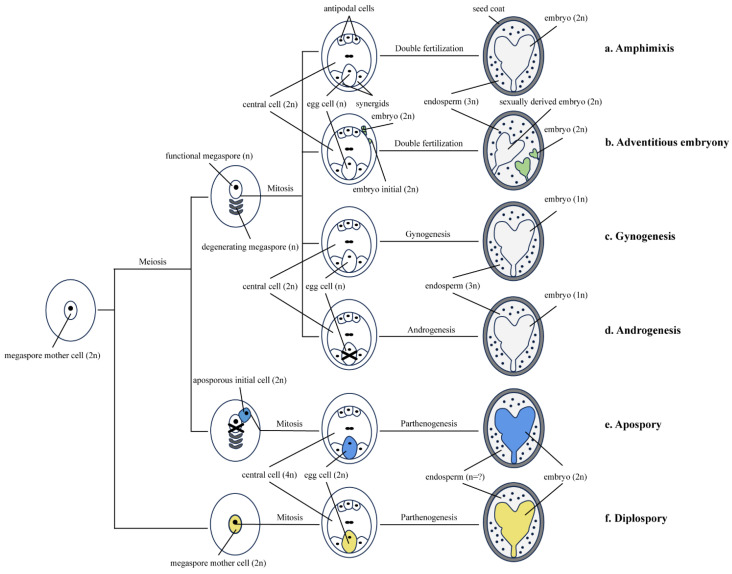
Comparison of amphimixis and apomixis. (**a**). Amphimixis is a process of seed reproduction in which the egg cell and polar nucleus produced by the macrospore mother cell through meiosis combine with the sperm formed by spermatocyte to form an embryo and endosperm; (**b**). Adventitious embryony is the direct production of adventitious embryos from somatic cells and then develops into plants; (**c**). Gynogenesis refers to the phenomenon that the egg cells develop directly into individuals without fertilization; (**d**). Androgenesis refers to the process of pollination in which the sperm enters the embryo sac without nuclear fusion, the egg nucleus disintegrates and disappears, and the sperm directly develops into an embryo; (**e**,**f**). The embryos produced by diplospory and apospory did not originate from the sporophyte of the ovary but from the egg cells of the unreduced embryo sac; (**e**). In apospory, the unreduced embryo sac is formed from a nucellus cell, which has the characteristic of non-spore initiation before mitosis, thus producing the unreduced embryo sac; (**f**). In diplospory, the unreduced embryo sac is developed from the macrospore mother cell without meiosis (Black cross: Degradation) (Modified from [3] and added new contents with Adobe Illustrator 2023).

**Figure 2 ijms-25-11378-f002:**
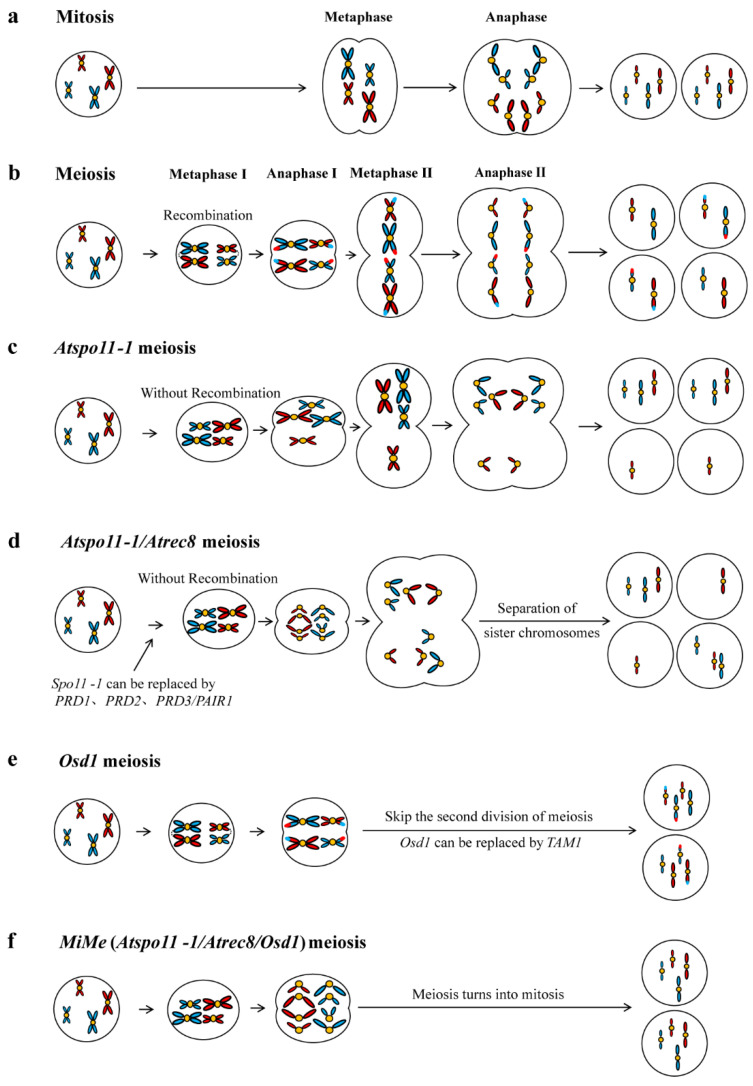
Pattern graph of *MiMe* mechanism. (**a**). Wild-type mitosis; (**b**). Wild type meiosis; (**c**). *Atspo11-1* mutant meiosis. Homologous recombination of chromosomes was inhibited during metaphase I and appeared to have unequal separation in anaphase I. Later, sterile gametes are formed after meiosis II; (**d**). *Atspo11-1/Atrec8* double mutant meiosis. Homologous recombination of chromosomes was inhibited, and sister chromatids were separated advanced during metaphase I; later, sterile gametes are formed after meiosis II; (**e**). *Atosd1* mutant meiosis. The second division of meiosis was skipped; (**f**). *Atspo11-1/Atrec8/Atosd1* triple mutant meiosis (*MiMe*). Meiosis is transformed into mitosis, and diploid gametes are produced (Modified from [27] with Adobe Illustrator 2023.).

**Figure 3 ijms-25-11378-f003:**
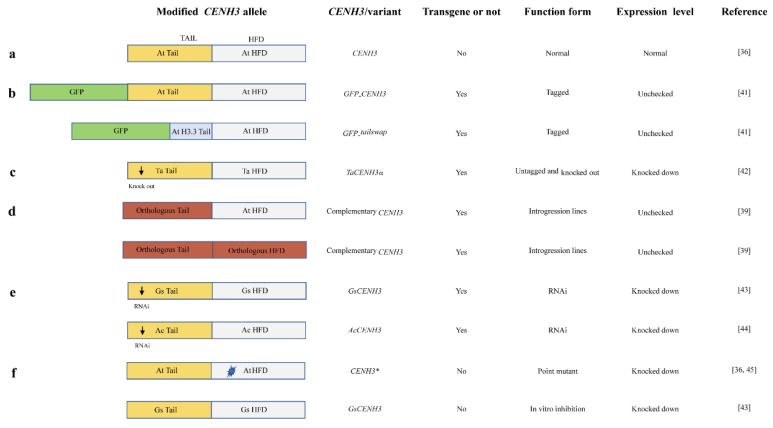
Haploid induction (genome elimination) schematic by altering the expression of native *CENH3*. (**a**). Centromere specific histone H3 variant CENH3 [34]; (**b**). GFP tagged at the N-terminal of CENH3 (called *GFP-CENH3*); Replacing the N-terminal tail domain of CENH3 with that from the H3.3 variant, and also tagged with GFP (called *GFP-tailswap*) [39]; (**c**). Obtaining *TaCENH3α*-heteroallelic combinations by knocking out untagged *CENH3* [40]; (**d**). Constructing introgression lines with competitive orthologous *CENH3* variants from other species [37]; (**e**). Mock sexual reproduction with RNAi [41,42]; (**f**). Triggering haploid plants by outcrossing with point mutant *CENH3* lines; Mock sexual reproduction with in vitro inhibition [34,41,43]. (H3: A highly conserved protein within eukaryotes; H3.3: A canonical histone; TAIL: N-terminal tail; HFD: C-terminal histone folding domain; GFP: Green fluorescent protein; Orthologous tail: Orthologous N-terminal tail; Orthologous HFD: Orthologous C-terminal histone folding domain; Black arrow: RNAi; *CENH3**: Transgene with point mutation.) (By Adobe Illustrator 2023).

**Figure 4 ijms-25-11378-f004:**
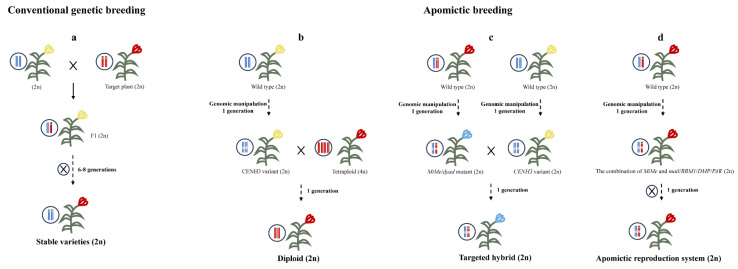
The difference between conventional genetic breeding and apomictic breeding. (**a**). Conventional genetic breeding generally takes 6–8 generations before stable line varieties for production are obtained; (**b**). Outcrossing the *CENH3* variant with tetraploid plants can result in diploid ones [39]; (**c**). Outcrossing the *MiMe*/*dyad* mutant with the *CENH3* variant can produce cloning seeds of the targeted hybrid with GOI [39]; (**d**). Combining *matl* (mutant) or overexpressing of *BBM1*/*DMP*/*PAR* with *MiMe* can obtain an apomictic reproduction system of hybrid [61,65,68,136]. (Yellow flower plant: A variety; Red flower plant: Another variety; Blue flower: Target trait; GM: Genomic manipulation; GOI: Gene of interest; Green locus in the genome: Targeted gene or locus; Gray locus in the genome: Genomic manipulation.) (By Adobe Illustrator 2023).

**Table 1 ijms-25-11378-t001:** Application of apomixis-related genes. *MiMe*: *Mitosis instead of Meiosis*; *SWI1*: *DYAD*/*SWITCH1*; *CENH3*: *CENTROMERIC HISTONE3*; *MATL*: *MATRILINEAL*; *PLA1*: *PHOSPHOLIPASE A1*; *NLD*: *NOT LIKE DAD*; *BBM1*: *BABY BOOM 1*; *ZmDMP*: *Zea mays DUF679 Domain Membrane Protein*; *PAR*: *PARTHENOGENESIS*; *RWP*: *RKD*, *RWP-RK domain-containing*.

Gene/Mutant		Molecular Function	Function in Apomixis	Species	References
*MiMe*	*SPO11-1* or *PRD1* or *PRD2* or *PRD3/PAIR1*	Blocks homologous chromosome pairing and recombination	Production of functional diploid gametes	*Arabidopsis*, rice	[26,27,28,29]
	*DIF1/SYN1/REC8*	Causes early separation of sister chromatids			[27]
	*OSD1* or *TAM*/*CYCA1;2*	Produces diploid gametes by omission the division of meiosis II			[27,30,31]
*SWITCH1/DYAD* (*SWI1*)		Encodes a nuclear entwining protein	Production of fertile undiminished female gametes	*Arabidopsis*	[32,33]
*CENH3*		Causes the elimination of chromosomes with mutant CENH3 protein and haploid induction	Haploid offspring induction	*Arabidopsis*, wheat, onion, cotton, barley	[34,35,36,37,38,39,40,41,42,43,44,45,46]
*MATL/ZMPLA1/NLD*		Contributes towards all pleiotropic defects associated with haploid induction	Parthenogenetic haploid offspring induction	Arabidopsis, maize, rice	[47,48,49,50,51]
*BABY BOOM 1*(*BBM1*)		Functions in cell proliferation, plant growth and development, and induces embryogenesis	Parthenogenetic haploid offspring induction	*Brassica napus* L., *Arabidopsis*, maize, rice	[52,53,54,55,56,57,58,59,60,61,62,63]
*ZmDMP*		Increases haploid induction rate (HIR) and the endosperm aborted kernels (EnAs)	Haploid offspring induction	*Arabidopsis*, maize	[64,65]
*PARTHENOGENESIS* (*PAR*)		Triggers embryo development in unfertilized egg cells	Egg cell division without fertilization induction	*Taraxacum officinale, Setaria italica*, rice	[66,67,68]
*RWP* (*RKD*, *RWP-RK domain-containing*)		Maintains egg-cell identity	Polyembryony induction	*Arabidopsis*, *Fortunella hindsii*, citrus, mango	[69,70,71,72]

**Table 2 ijms-25-11378-t002:** Epigenetic genes associated with apomixis. *DMT102*: *DNA methyltransferase102*; *DMT103*: *DNA methyltransferase103*; *FIE*: *FERTILIZATION-INDEPENDENT ENDOSPERM*; *AGO9*: *ARGONAUTE9*; *AGO104*: *ARGONAUTE104*; *SWI1*: *DYAD/SWITCH1*; *ORC*: *ORIGIN RECOGNITION COMPLEX*; *GID1*: *GIBBERELLIN-INSENSITIVE DWARF1*; *MSP1*: *MULTIPLE SPOROCYTE1*; *SERK*: *SOMATIC EMBRYOGENESIS RECEPTOR-LIKE KINASE*; *MSP1*: *MULTIPLE SPOROCYTE1*.

Gene	Species	Affected Process	Gene Function	Reference
*DMT102* and *DMT103*	*Maize-Tripsacum hybrid*	Differentiation between apomictic and sexual reproduction	Produces a phenotype similar to apomixis	[84]
*FIE*	*Malus hupehensis, Solanum lycopersicum*	Autonomous endosperm	Fertilization independent endosperm	[81,83]
*AGO9*	*Arabidopsis*	Diplospory	Controls female gamete formation by limiting the specification of gametophyte precursors in a noncell autonomous manner	[85]
*AGO104*	*Tripsacum*	Diplospory	Catalytic the component of RNA-induced protein complex of gene silencing	[86]
*SWITCH1/DYAD (SWI1)*	*Arabidopsis*	Diplospory	Meiosis specific chromatin associated protein	[87]
*ORC*	*Paspalum simplex*	Adventitious embryony	Controls DNA replication and cell differentiation in eukaryotes	[88]
*GID1*	*Brachiaria brizantha*	Diplospory	Involved in the differentiation of single megaspore mother cells in ovule development	[62]
*MSP1*	*Oryza sativa*	Adventitious embryony	Controls early sporogenic development	[89]
*SERK*	*Poa pratensis*, *Paspalum notatum*	Adventitious embryony	Plays a key role in embryo sac development	[7]
*APORSTART*	*Medicago sativa*	Diplospory	Speculated to participate in the formation of 2n eggs in apomixis	[77]

## Data Availability

All data supporting the findings of this study are available in the paper and online.

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
