# Peer review of "Recent Progress on Plant Apomixis for Genetic Improvement"

_ijms, 2024, doi:10.3390/ijms252111378_

Round 1
Reviewer 1 Report
Comments and Suggestions for Authors
The manuscript titled ‘Recent Progress on Plant Apomixis for Genetic Improvement’ by Xue et al. provides a comprehensive review of apomixis and its applications in the genetic improvement of plants. It extensively details probable gene functions and the key factors regulating apomixis and outlines various technological advancements that have enhanced the understanding and application of this process. The authors cite relevant examples of how apomixis has been applied in plant breeding.
However, despite its merits, the manuscript's composition does not meet the journal's standards. It suffers from a lack of flow and clarity, making it difficult to follow. Additionally, the text is riddled with grammatical errors, redundancies, incomplete statements, and informal language. I recommend that the authors reorganize the manuscript to significantly improve its readability. For detailed comments and suggestions, please refer to the attached PDF file.

As I mentioned, the text is rife with grammatical errors, redundancies, incomplete statements, and informal language. I recommend that the authors reorganize the manuscript to improve its readability significantly.
Author Response
Comment: The manuscript titled ‘Recent Progress on Plant Apomixis for Genetic Improvement’ by Xue et al. provides a comprehensive review of apomixis and its applications in the genetic improvement of plants. It extensively details probable gene functions and the key factors regulating apomixis and outlines various technological advancements that have enhanced the understanding and application of this process. The authors cite relevant examples of how apomixis has been applied in plant breeding.
However, despite its merits, the manuscript's composition does not meet the journal's standards. It suffers from a lack of flow and clarity, making it difficult to follow. Additionally, the text is riddled with grammatical errors, redundancies, incomplete statements, and informal language. I recommend that the authors reorganize the manuscript to significantly improve its readability. For detailed comments and suggestions, please refer to the attached PDF file.
Response: Thank you very much for your careful comments and suggestions to this manuscript.
We have made every effort to improve this manuscript, including adding a section on plant genomic evolution, adjusting the logicality and readability of the manuscript, and standardizing the language. It is hopeful that this manuscript can provide more references for the future development of apomictic breeding. We have uploaded the modified file based on the PDF you attached, named ijms-3226656-response 1. In addition, this manuscript has been polished by a native English speaker, making the language of this manuscript more scientific and readable. The certificate has been uploaded as an attachment.
Thanks again for your contribution to this manuscript.

Reviewer 2 Report
Comments and Suggestions for Authors
The manuscript is well-written and presents relevant information on the topic. Minor suggestions were made. We request that figures and tables be sent for a complete work analysis.
This work presents the application of apomixis molecular mechanisms in plant genetic improvement.
Line 19: “commonly “, recommends using words that convey a more affirmative idea than what will be addressed in the review.
The citation numbering is in alphabetical order, is that correct?
Line 28: There is no figure in the document
Line 28-30: Review the wording of the sentence, confusing information
Line 54: Explain.
Line 70: Put in the order of the items described.
Line 160: The table is not in the file.
Line 166: D'Erfurth et al: Number?
Line 217 and 224: Ravi & Chan: number?
Author Response
Comment: The manuscript is well-written and presents relevant information on the topic. Minor suggestions were made. We request that figures and tables be sent for a complete work analysis.
This work presents the application of apomixis molecular mechanisms in plant genetic improvement.
Line 19: “commonly “, recommends using words that convey a more affirmative idea than what will be addressed in the review.
The citation numbering is in alphabetical order, is that correct?
Line 28: There is no figure in the document.
Line 28-30: Review the wording of the sentence, confusing information.
Line 54: Explain.
Line 70: Put in the order of the items described.
Line 160: The table is not in the file.
Line 166: D'Erfurth et al: Number?
Line 217 and 224: Ravi & Chan: number?
Response: Thank you for your comments and affirmation. We have uploaded the figures and tables again and have responded to your suggestions as follows.
Comment 1: Line 19: “commonly “, recommends using words that convey a more affirmative idea than what will be addressed in the review.
Response 1: Thanks for your advice, “commonly” has been deleted.
Comment 2: The citation numbering is in alphabetical order, is that correct?
Response 2: Yes, and we have rearranged the citation numbers in the order they appeared in the revised manuscript. Thanks again.
Comment 3: Line 28: There is no figure in the document.
Response 3: We have uploaded the figures and tables again following the main manuscript, and attached higher resolution figures and tables in the revised manuscript.
Comment 4: Line 28-30: Review the wording of the sentence, confusing information.
Response 4: Thank you for your kind opinion. We could not be more agreeable with the comment and have rewritten the sentence in the revised manuscript as the following: Seed formation also occurs in asexual reproduction, where the normal process involving meiosis and fertilization is replaced. Agamogenesis refers to a mode that does not rely on gametogamy to produce seeds.
Comment 5: Line 54: Explain.
Response 5: Thanks for your advice. This sentence is meant to emphasize the importance of the regulatory molecular mechanism of asexual reproductive development in plants.
Comment 6: Line 70: Put in the order of the items described.
Response 6: Thank you for your kind comment. This part has been adjusted appropriately with logicality.
Comment 7: The table is not in the file.
Response 7: We have uploaded the figures and tables again following the main manuscript, and attached higher resolution figures and tables in the revised manuscript.
Comment 8: D'Erfurth et al: Number?
Response 8: The citation number is 29, and all the references have been carefully confirmed accordingly.
Comment 9: Line 217 and 224: Ravi & Chan: number?
Response 9: The citation number is 42, and all the references have been carefully confirmed accordingly.
Thanks for your professional comments. We have made every effort to improve this manuscript, including adding a section on plant genomic evolution, adjusting the logicality and readability of the manuscript, and standardizing the language. The English language editing certificate has been uploaded as an attachment. It is hopeful that this manuscript can provide more references for the future development of apomictic breeding.
Thanks again for your contribution to this manuscript.

Reviewer 3 Report
Comments and Suggestions for Authors
The manuscript titled "Recent Progress on Plant Apomixis for Genetic Improvement" is considered as one of the most interesting review articles which deal with the plant genome analysis. The manuscript discussed many items and used many tools for examining the mechanism of the molecular plant apomixis and the genetic improvement. But i hope if the authors make a good discussion about epigenetics and plant genome evolution. Also, the repetitive DNA, transposon elements and their roles in the plant genome evolution. As well. The recent citation also missing in the manuscript and the headlines along the manuscript should be more precise.
Author Response
Comment: The manuscript titled "Recent Progress on Plant Apomixis for Genetic Improvement" is considered as one of the most interesting review manuscripts which deal with the plant genome analysis. The manuscript discussed many items and used many tools for examining the mechanism of the molecular plant apomixis and the genetic improvement. But i hope if the authors make a good discussion about epigenetics and plant genome evolution. Also, the repetitive DNA, transposon elements and their roles in the plant genome evolution. As well. The recent citation also missing in the manuscript and the headlines along the manuscript should be more precise.
Response: We greatly appreciate your professional comments on this review.
As is well known, epigenetics and plant genomic evolution are important for apomixis. In order to make this manuscript more comprehensive, we have referred to more relevant literature and added section 4.4 “plant genomic evolution” and made logical adjustments to the manuscript. In addition, this manuscript has been polished by a native English speaker, making the language of this manuscript more scientific and readable. The certificate has been uploaded as an attachment. Finally, we have cited several recent relevant manuscripts and modified the subtitle to make the manuscript more standardized.
Thanks again for your contribution to this manuscript.

Reviewer 4 Report
Comments and Suggestions for Authors
Kindly follow my suggestions and corrections addressed in my report.

Moderately English Language editing is needed.
Author Response
Comment 1: - 'which has great potential in the reservation of genes of interest and fixation of heterosis' could be clearer.
- 'commonly focuses' is redundant.
- Simplify sentences like 'shed light on the decipherment and application' to improve readability.
- The sentence 'Understanding and utilizing the molecular mechanisms...' needs clearer phrasing
Response 1: Thanks for your advice. According to your opinions, we have made modifications to these sentences in the Abstract section after reading through the full text to make them more readable.
Comment 2: - 'Seed formation also occurs in asexual reproduction...' needs clarity, use 'where' instead of 'while'.
- Avoid excessive use of technical terms, especially when describing the difference between sexual and asexual reproduction.
- Simplify the sentence 'Seed production of asexual reproduction is considered the Holy Grail...' for a formal tone.
Response 2: Thank you for your advice on the language specification of the manuscript. We have carefully read and revised this section. In order to enhance language diversity, we have replaced some “sexual reproduction” and “asexual reproduction” with “gamogenesis” and “agamogenesis”, and simplified “Seed production of asexual reproduction is considered the Holy Grail...” as “Asexual seed production is considered the Holy Grail of plant biology” in the revised manuscript.
Comment 3: - Correct 'apomixes' to 'apomixis'.
- Avoid redundant terms like 'female parthenogenesis and male parthenogenesis...'
- Clarify complex terms and definitions, particularly in the sections on adventitious embryo propagation.
Response 3: Thank you for your kind advice. We have corrected the wrong words and made modifications to the 2.3 “adventitious embryony” part. In addition, by reading relevant article, we have changed “female parthenogenesis” and “male parthenogenesis” to “Gynogenesis” and “androgenesis”.
Comment 4: - Repetition of 'sexual' should be minimized.
- Some explanations on SWI1 and MiMe lack clarity and require better transition.
- Remove excessive punctuation or rephrase longer sentences for clarity.
Response 4: Thank you for your professional advice, we highly agree with your comments. We have made revisions and language polishing to this section, hoping to increase the logicality and readability of this manuscript.
Comment 5: - Avoid redundant phrases like 'as previously stated'.
- The epigenetics section is dense; consider breaking it into smaller, more digestible parts.
- Clarify hormone-related mechanisms by specifying the role of cytokinin and auxin.
Response 5: Thank you for your comments. Due to the importance of language norms and the significant impact of hormones on apomixis, we have deleted “as previously stated”, logically adjusted the Part 4. “Main affecting factors of apomixis” and clarified the role of cytokinin and auxin” to make this manuscript more scientific and readable.
Comment 6: - The phrase 'fix the phenotype of Hieracium pilosella...' could use more clarity. - Avoid repetition of terms like 'in vivo HI'; ensure HI is defined clearly the first time it's mentioned.
- Clarify the link between haploid induction and CRISPR in the HI-Edit section.
Response 6: Thank you for your comments. HI-Edit combines haploid induction (HI) with CRISPR/Cas9 gene-editing (Edit) technology to directly achieve haploid induction with improved crop breeding through the genomic editing of specific agronomic loci at the same time. HI-Edit can not only rapidly introduce MiMe phenotypes or haploid induction types using the modification of apomixis-related genes in hybrid plants, but also genes of interest can also be edited by CRISPR/Cas9 gene-editing simultaneously, thus greatly promoting the application progress of personalized breeding for apomixis. According to your opinions, we have carefully reviewed and polished the full text about this problem in order to make the manuscript more logical and easily understood.
Comment 7: - Split long sentences, such as 'A haploid induction (HI) line can induce haploid production...' for better readability.
- Ensure consistency when discussing quantitative trait loci (QTL) for easier understanding.
- Technical details regarding Brassica napus haploids should be broken into smaller parts.
Response 7: Thank you for your comments. We have revised this section based on your suggestions to ensure the scientificity, logicality and readability of this manuscript.
Comment 8: - Explain 'HI-Edit' clearly when introducing it.
- Improve clarity in the CRISPR discussion by specifying why it's significant for haploid breeding.
- Simplify vague statements in the final section to ensure all research outcomes are clearly understood.
Response 8: Thank you for your comments. Apomictic genes can be utilized for haploid induction through genetic editing, overexpression, and other transgenic strategies. HI-Edit can not only rapidly introduce MiMe phenotypes or haploid induction types using the modification of apomixis-related genes in hybrid plants, but also genes of interest can also be edited by CRISPR/Cas9 gene-editing simultaneously, thus greatly promoting the application progress of personalized breeding for apomixis. Besides, we have simplified the vague statements in the last section to ensure that readers can better understand this manuscript.
Finally, we have cited several recent relevant manuscripts and modified the subtitle to make the manuscript more standardized. In addition, this manuscript has been polished by a native English speaker, making the language of this manuscript more scientific and readable. The certificate has been uploaded as an attachment.
Thanks again for your contribution to this manuscript.

Round 2
Reviewer 1 Report
Comments and Suggestions for Authors
The authors have addressed all my questions and suggestions. The revised manuscript now exhibits polished grammar and significantly improved readability. However, I would like to point out one error. While the authors have replaced 'female parthenogenesis' and 'male parthenogenesis' with 'gynogenesis' and 'androgenesis' respectively in the main text, Figure 1 has not been updated to reflect these changes. It is crucial to modify the figure to align with the main text to prevent any confusion or inconsistencies.
Aside from the issue mentioned, I found no major concerns with the manuscript. I recommend its publication in its current form.
Reviewer 4 Report
Comments and Suggestions for Authors
Some minor English errors needs your attention.
Comments on the Quality of English LanguageMinor revisions